# Fungi found in Mediterranean and North Sea sponges: how specific are they?

Mohd Azrul Naim[1,2], Hauke Smidt[1] and Detmer Sipkema[1]

[1] Laboratory of Microbiology, Wageningen University, Wageningen, The Netherlands
[2] Department of Biotechnology, International Islamic University, Jalan Istana, Malaysia

## ABSTRACT

Fungi and other eukaryotes represent one of the last frontiers of microbial diversity in the sponge holobiont. In this study we employed pyrosequencing of 18S ribosomal RNA gene amplicons containing the V7 and V8 hypervariable regions to explore the fungal diversity of seven sponge species from the North Sea and the Mediterranean Sea. For most sponges, fungi were present at a low relative abundance averaging 0.75% of the 18S rRNA gene reads. In total, 44 fungal OTUs (operational taxonomic units) were detected in sponges, and 28 of these OTUs were also found in seawater. Twenty-two of the sponge-associated OTUs were identified as yeasts (mainly *Malasseziales*), representing 84% of the fungal reads. Several OTUs were related to fungal sequences previously retrieved from other sponges, but all OTUs were also related to fungi from other biological sources, such as seawater, sediments, lakes and anaerobic digesters. Therefore our data, supported by currently available data, point in the direction of mostly accidental presence of fungi in sponges and do not support the existence of a sponge-specific fungal community.

Subjects Marine Biology, Microbiology, Mycology
Keywords Marine sponge, Fungi, Yeast, *Malasseziales*, Symbiosis

# INTRODUCTION

Fungi constitute a large proportion of microbial diversity on Earth (*Hawksworth, 2001*; *Mueller & Schmit, 2007*) and are considered key players in terrestrial environments for decomposition of organic matter, nutrient recycling or as symbionts of plants or other fungi by improving host fitness (*Rodriguez, Redman & Henson, 2004*). Global fungal richness has been estimated between 1.5 and 1.6 million species (*Hawksworth, 1991*; *Hawksworth, 2001*), but despite extensive attempts to study and characterize fungi, their diversity remains underexplored. Most of our knowledge about evolution and ecology of fungi has been derived from cultured representatives of fungi from the terrestrial environment. In comparison, much less is known about marine fungal diversity and ecology.

Marine fungi belong to a wide variety of families, but appear to be present only in low numbers (compared to bacteria) in seawater and have been estimated to contribute up to only 0.6% of the global fungal richness (*Richards et al., 2012*; *Richards et al., 2015*). The generally accepted definition of a marine fungus is broad and is based on the habitat as described by *Kohlmeyer & Volkmann-Kohlmeyer (1990)*: "obligate marine fungi are those that grow and sporulate exclusively in a marine or estuarine habitat; facultative marine

Corresponding author
Detmer Sipkema,
detmer.sipkema@wur.nl

fungi are those from freshwater and terrestrial milieus able to grow and possibly also sporulate in the marine environment''. Fungi are considered to play a role in marine ecosystems as saprotrophs, parasites, or symbionts (*Hyde et al., 1998*). Different habitats of marine fungi have been studied including deep-sea sediments (*Singh et al., 2010*), hydrothermal vents (*Le Calvez et al., 2009*), seawater (*Kis-Papo et al., 2003*) and anoxic regions in the deeper parts of the oceans (*Bass et al., 2007*). Marine fungi have also been described to be associated to marine animals, such as sea fans (*Toledo-Hernández et al., 2008*), corals (*Amend, Barshis & Oliver, 2012*; *Bentis, Kaufman & Golubic, 2000*) and algae (*Loque et al., 2009*).

Marine sponges provide yet another habitat for fungi, but whereas bacterial and archaeal diversity in sponges has been thoroughly characterized (*Simister et al., 2012*; *Taylor et al., 2007*), knowledge of sponge-associated fungal diversity remains scarce (*Webster & Taylor, 2012*). Indirect evidence of interactions between marine sponges and fungi was provided by the detection of fungal introns in the genomes of some marine sponge species that were most probably acquired by horizontal gene transfer (*Rot et al., 2006*). Fungi have been repeatedly isolated from many sponge species (*Baker et al., 2009*; *Höller et al., 2000*; *Liu et al., 2010*; *Passarini et al., 2013*; *Paz et al., 2010*; *Pivkin et al., 2006*; *Wang, Li & Zhu, 2007*; *Wiese et al., 2011*). Despite a gap of knowledge about the fungal life cycle in sponges and other environmental fungi (*Richards et al., 2012*), it is enticing to speculate about the role of sponge-associated fungi. Many sponge-derived fungi have been found to produce molecules with antimicrobial activity and may be involved in chemical protection of their hosts (*Indraningrat, Smidt & Sipkema, 2016*). In addition, they are potentially parasites or pathogens to sponges themselves or sponges may solely serve as a reservoir for (pathogenic) marine fungi once they (or their spores) are trapped by the sponge through its efficient water filtration system. For instance, *Metchniskowia* spp., which were found in *H. simulans* and Antarctic sponges (*Baker et al., 2009*; *Vaca et al., 2013*), are known to be responsible for the infection and mortality of prawns (*Chen et al., 2007*) and *Aspergillus sydowii*, a pathogen of sea fans was also isolated from the marine sponge *Spongia obscura* (*Ein-Gil et al., 2009*). However, the true diversity of sponge-associated fungi has until recently been difficult to establish (*Schippers et al., 2012*). The reason is that designing specific PCR primers for universal fungal phylogenetic marker genes remains challenging since sponges and fungi are closely related from an evolutionary perspective (*Borchiellini et al., 1998*). This was demonstrated in a Hawaiian sponge study where many sponge-derived sequences were found in clone libraries generated from PCR amplicons using 'fungi-specific' 18S rRNA gene and ITS primers (*Gao et al., 2008*; *Jin et al., 2014*). With the use of next-generation sequencing, it is now possible to overcome such interference of the sponge host phylogenetic marker genes by the sheer number of reads that is generated as shown in a number of recent studies (*He et al., 2014*; *Passarini et al., 2015*; *Rodríguez-Marconi et al., 2015*; *Wang et al., 2014*).

The aim of this study was to assess the diversity and specificity of sponge-associated fungi. Seven shallow water sponge species from two different regions, the North Sea and the Mediterranean Sea, were sampled to identify host specificity of the associated fungal communities and the impact of geography on community structuring.

## MATERIALS AND METHODS

### Sample collection and processing

The North Sea sponges *Halichondria panicea* (P1–P3), *Haliclona xena* (X1–X3), and *Suberites massa* (M1–M3) were collected on December 3rd, 2008, from the Oosterschelde estuary, at the dive site Lokkersnol (N 51°38′58.07″, E 3°53′5.11″) by SCUBA diving at a depth of approximately 14 m. Sponge collection of the North Sea sponges was approved by the Provincie Zeeland (document number 0501560). The Mediterranean sponges *Aplysina aerophoba* (A1–A3), *Petrosia ficiformis* (F1–F3), *Axinella damicornis* (D1–D3) *and Axinella verrucosa* (V1–V3) were collected by SCUBA diving offshore L'Escala, Spain (N 42°06′52.20″, E 03°10′06.52″) at a depth of approximately 15 m on January 15th, 2012. The collection of Mediterranean sponge samples was conducted in strict accordance with Spanish and European regulations within the rules of the Spanish National Research Council with the approval of the Directorate of Research of the Spanish Government. Initial identification of sponges based on their morphology was done by Prof. Rene Wijfels and Dr. Klaske Schippers for the North Sea sponges and by Dr. Detmer Sipkema and Prof. MJ Uriz for the Mediterranean sponges. All sponge specimens from the North Sea and the Mediterranean Sea were collected in triplicate. Specimens were brought to the surface in ziplock plastic bags and were immediately transported to the laboratory in excess of seawater and processed. Each sponge specimen was separately submerged and cut into pieces of approximately 0.5 ml that contained both pinacoderm and choanoderm and was rinsed three times in a large volume of autoclaved artificial seawater (26.52 g NaCl, 2.45 g $MgCl_2$, 0.73 g KCl, 1.14 g $CaCl_2$ and 3.31 g $MgSO_4$/l) and kept at $-80$ °C until further processing. Furthermore, between 5 and 10 l of seawater from both locations was collected and filtered immediately upon collection onto a 0.2 μm polycarbonate filter with a diameter of 47 mm (GE Osmonics, Minnetonka, MN, USA). Each filter was then stored in a sterile 15 ml Falcon tube and kept at $-80$ °C until further processing.

### DNA extraction and PCR amplification

Total DNA was extracted from North Sea sponges using the DNeasy Blood & Tissue Kit (Qiagen, Hilden, Germany) according to the tissue extraction protocol. For Mediterranean sponges total DNA was extracted using the FastDNA SPIN kit for soil (MP Biomedicals, Solon, OH, USA) with the aid of a PreCellys® homogenizer (Bertin Technologies, France) following the manufacturers' protocol. For seawater samples, filters were cut in two pieces, and DNA was extracted from half of the filter with the FastDNA SPIN kit for soil following the same protocol that was used for the Mediterranean sponges.

Amplification of partial 18S rRNA genes was performed using the GoTaq® Hot Start Polymerase kit (Promega GmbH, Mannheim, Germany) with the universal fungal primers FF390 (CGATAACGAACGAGACCT) and FR1 (ANCCATTCAATCGGTANT) (*Vainio & Hantula, 2000*), which amplify a region of approximately 350 base pairs that includes the V7 and V8 hypervariable regions of the eukaryotic small-subunit rRNA gene. Sample-specific barcodes and adapter sequences were added to the forward primer as described previously (*Hamady et al., 2008*) (Table S1). The PCR conditions were: initial denaturation (2 min at 95 °C) followed by 30 cycles of denaturation (30 s at 95 °C), primer annealing (45 s at
50 °C), primer extension (60 s at 72 °C), and a final extension (10 min at 72 °C). The final PCR mixture (50 μl) contained 1 × GoTaq® Green Flexi buffer, 1.5 mM MgCl$_2$, 0.2 mM of each dNTP, 0.2 μM of each primer, 1.25 U GoTaq® Hot Start Polymerase and 10 ng template DNA. PCR reactions were carried out in triplicate, pooled and cleaned using the High Pure PCR Cleanup Micro Kit (Roche Diagnostics GmbH, Mannheim, Germany). Purified DNA concentrations were measured with a Qubit® 2.0 Fluorometer (Invitrogen, Carlsbad, CA, USA). An equimolar mixture with a final concentration of 1 μg/ml PCR product was prepared, electrophoresed on 1.25% (w/v) agarose gel and subsequently purified using the Milipore DNA Gel Extraction Kit (Milipore, Billerica, MA, USA). The pooled purified DNA was pyrosequenced on a 454 Roche platform at GATC Biotech, Konstanz, Germany. Pyrosequencing data was deposited at the European Bioinformatics Institute with accession numbers ERS225550–ERS225575.

## Sequence analysis

Pyrosequencing data was analysed using the QIIME pipeline v1.5.0 (*Caporaso et al., 2010b*). Low quality sequences were removed using default parameters, including: (i) reads with fewer than 200 or more than 1,000 nucleotides; (ii) reads with more than six ambiguous nucleotides, (iii) homopolymer runs exceeding six nucleotides, (iv) reads with missing quality scores and reads with a mean quality score lower than 25, and (v) reads with mismatches in the primer sequence. Operational taxonomic units (OTUs) were identified at the 97% identity level using UCLUST v1.2.22 embedded in QIIME (*Edgar, 2010*). Representative sequences from the OTUs were aligned using PyNAST (*Caporaso et al., 2010a*) against the aligned SILVA 104 core set. Taxonomic assignment of all OTUs was performed using the BLAST algorithm against the QIIME-compatible version of the SILVA 104 release (*Pruesse et al., 2007*) as reference database. Possible chimeric OTUs were identified using QIIME's ChimeraSlayer and removed from the initially generated OTU list, producing a final set of 585 non-chimeric OTUs.

Since the majority of reads and OTUs were not of fungal origin, these were removed from the dataset prior to further analysis. The fungal OTU matrix from sponge and seawater samples was used to calculate species richness estimates using the 'Species Frequency/Abundance Data' option with default settings with the SPADE program (*Chao & Shen, 2013*). Fungal community coverage was estimated using Good's coverage in which Coverage = 1–(number of singleton OTUs/number of reads). In addition, the fungal OTU matrix was used for betadiversity analysis and samples that had <10 fungal reads were excluded from this analysis. Bray Curtis dissimilarity was calculated based on square root transformed relative abundance data and on presence-absence. Principal coordinates analysis (PCoA) was performed to represent the samples in a low dimensional space. All statistical analyses were performed using the multivariate statistical software package Primer V7 (Primer-E Ltd., Plymouth, UK).

## Phylogenetic analysis of sponge-associated fungi

For a more detailed phylogenetic analysis of the fungal OTUs from sponges we selected all fungal OTUs that were found in sponge samples excluding only singletons. Representative

reads for these OTUs were deposited at NCBI genbank with accession numbers MF094397–MF094440 (also available in Table S4). These representative sequences were aligned using the SILVA online SINA alignment service (*Pruesse et al., 2007*). Each OTU was complemented with the two most closely related 18S rRNA gene sequences as determined by a BLAST search against the NCBI nucleotide database. Nearest neighbour sequences and published 18S rRNA sponge-derived fungal sequences longer than 700 nucleotides (*Baker et al., 2009*; *Simister et al., 2012*) were downloaded from the SILVA database (release 108) and together with aligned OTUs from our own dataset imported into the ARB software package (*Ludwig et al., 2004*). Nearest neighbour sequences and published sponge-derived fungal 18S rRNA sequences were first used to construct a Bayesian phylogenetic tree. Ambiguous regions of the alignment were systematically removed using the program Gblocks v.0.91b (*Castresana, 2000*). The default program parameters were used, except allowing a minimum block length of three and gaps in 50% of positions. Phylogenetic trees were created by Bayesian analysis, using MrBayes v3.2 (*Ronquist et al., 2012*) at the freely available Bioportal server (http://www.bioportal.uio.no). All parameters were treated as unknown variables with uniform prior-probability densities at the beginning of each run, and their values were estimated from the data during the analysis. All Bayesian analyses were initiated with random starting trees and were run for $10^7$ generations. The number of chains was set to four and Markov chains were sampled every 1,000 iterations. Points prior to convergence were determined graphically and discarded. Calculated trees were imported into ARB and short sequences obtained in this study were subsequently added by use of the ARB parsimony method without changing the tree topology.

**Molecular identification of sponge samples**

To verify the identity of the sponge species, the six OTUs that represented the highest number of reads per sponge species (i.e., OTU65: *S. massa*, OTU190: *H. xena*, OTU319: *A. aerophoba*, OTU320: *P. ficiformis*, OTU333: *Axinella damicornis* and *Axinella verrucosa* and OTU495: *H. panicea*) were compared with the non-redundant nucleotide database using the Blastn query (Table S2). Sanger reads larger than 900 nucleotides of the 18S rRNA gene amplicons of *H. panicea* (P1–P3) and *H. xena* (X1–X3) were published in another study (*Naim et al., 2014*) with accession numbers KC899022–KC899040. The 18S rRNA gene regions of the amplicons generated here do not overlap with the sequences KC899022–KC899040.

## RESULTS

After DNA sequence quality filtering, a total of 350,341 partial 18S rRNA non-chimeric reads were retained. Thirty five reads could not be classified at the domain level and were removed from the dataset prior to further analysis. The remaining reads clustered into 585 OTUs. In total, 330,884 sequences (107 OTUs), contributing to 94.4% of all reads, were derived from the phylum *Porifera* (sponges). When only the sponge samples (and not seawater) were included, the percentage of sequences identified as *Porifera* increased to 96.8%. *P. ficiformis* was an outlier compared to the other sponge species as more than half of the reads obtained was classified as "non-sponge" (Fig. 1). Fungi represented 13.3% of

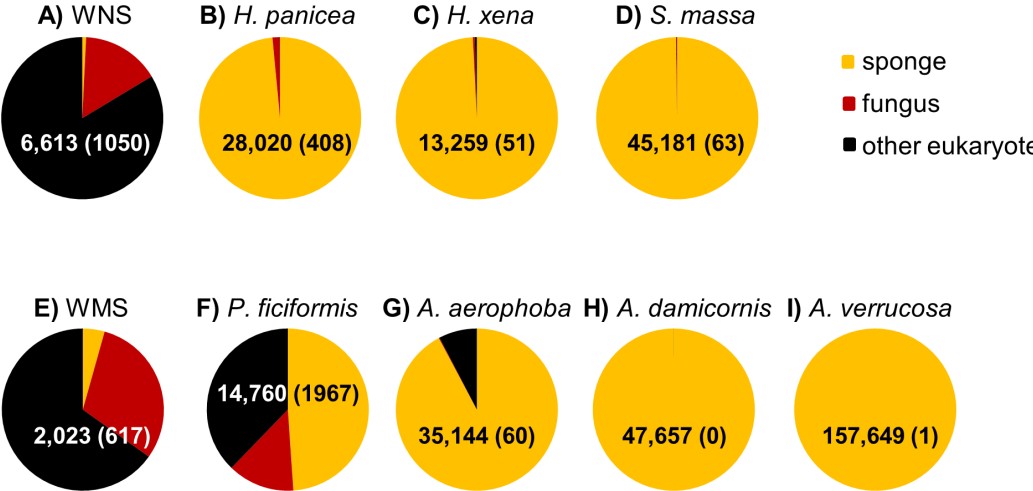

**Figure 1** **Relative abundance of sponge, fungal and "other eukaryotic" 18S rRNA gene sequences for sample types from the North Sea (A–D) and Mediterranean Sea (E–I).** The triplicate samples per sponge species are pooled and the numbers within the pie diagrams represent the number of sequences obtained for the sample type. The numbers in parentheses indicate the number of fungal reads obtained for each sample type. WNS, North Sea water; WMS, Mediterranean seawater.

the reads (1,967 fungal reads) in *P. ficiformis,* and other eukaryotes (defined as eukaryotes that are not sponge and not fungi) comprised 37.8% of the 18S rRNA gene reads. *A. aerophoba* also contained a substantial fraction of non-sponge reads (7.8%), but fungi represented only 0.17% (60 reads) of the 18S rRNA gene reads in this sponge species. In the North Sea sponge *H. panicea* fungi represented 1.5% (408 reads) of the 18S rRNA gene reads, while nearly no "other eukaryotic" sequences were found. For the other sponge species studied here the large majority of sequences obtained were classified as "sponge", and lower numbers of fungal reads were obtained: 63 for *S. massa*, 51 for *H. xena*, 1 for *A. verrucosa* and no fungal reads were found in *A. damicornis*. Generally, the three replicates per species yielded similar numbers of fungal reads (Table 1).

After removal of the 107 *Porifera* OTUs and 369 OTUs that were classified as "other eukaryotes" from the dataset, the remaining 109 fungal OTUs were used for diversity analyses. Highest fungal richness (Chao1) among the sponge samples was predicted for *P. ficiformis* and *H. panicea*, and richness estimates were generally positively correlated with the number of fungal reads obtained for the sponge samples (Table 1). Although some stratification of the fungal communities can be observed by sponge host species, no obvious discrimination based on geographical regions can be made (Fig. S1). The fungal OTUs detected in sponges belonged to the phyla *Ascomycota* and *Basidiomycota*, and four fungal-like OTUs were categorized in the environmental clade LKM11 (Figs. 2 and 3). OTUs belonging to the phylum *Chytridiomycota* were only found in seawater.

For a deeper phylogenetic analysis, OTUs obtained as singletons and those that were only found in the seawater samples were disregarded, leaving a final set of 44 fungal OTUs detected in sponges. Twenty-eight of these OTUs were also retrieved from seawater samples. Half of the 44 retained OTUs was identified as yeast (Fig. 4), representing 84% of the fungal

**Table 1** Number of reads, number of fungal reads, observed fungal OTUs, expected fungal OTUs (Chao1) and Coverage (Good's coverage) in seawater and sponge samples at a 97% sequence similarity threshold.

| Sample name | Abbrev. | Sponge taxonomic order | Total no. of filtered reads | No. of fungal reads | Fungal reads | | |
|---|---|---|---|---|---|---|---|
| | | | | | Observed OTUs | Expected OTUs | Coverage |
| North Sea water | WNS | | 6,613 | 1,050 | 69 | 94 ± 13 | 0.97 |
| *H. panicea* 1 | P1 | *Halichondrida* | 12,418 | 3 | 3 | 6 ± 4 | 0 |
| *H. panicea* 2 | P2 | *Halichondrida* | 8,389 | 149 | 15 | 29 ± 13 | 0.95 |
| *H. panicea* 3 | P3 | *Halichondrida* | 7,214 | 256 | 16 | 18 ± 3 | 0.98 |
| *H. xena* 1 | X1 | *Haplosclerida* | 3,728 | 22 | 5 | 6 ± 2 | 0.91 |
| *H. xena* 2 | X2 | *Haplosclerida* | 4,155 | 14 | 5 | 7 ± 3 | 0.79 |
| *H. xena* 3 | X3 | *Haplosclerida* | 5,381 | 15 | 5 | 5 ± 1 | 0.87 |
| *S. massa* 1 | M1 | *Hadromerida* | 14,265 | 25 | 8 | 18 ± 10 | 0.8 |
| *S. massa* 2 | M2 | *Hadromerida* | 19,955 | 27 | 9 | 17 ± 8 | 0.78 |
| *S. massa* 3 | M3 | *Hadromerida* | 10,961 | 11 | 9 | 16 ± 7 | 0.36 |
| Med. Sea water | WMS | | 2,023 | 617 | 17 | 17 ± 1 | 1.00 |
| *P. ficiformis* 1 | F1 | *Haplosclerida* | 5,704 | 1,200 | 23 | 29 ± 7 | 1.00 |
| *P. ficiformis* 2 | F2 | *Haplosclerida* | 4,623 | 486 | 17 | 17 ± 0 | 1.00 |
| *P. ficiformis* 3 | F3 | *Haplosclerida* | 4,462 | 281 | 16 | 16 ± 0 | 1.00 |
| *A. aerophoba* 1 | A1 | *Verongida* | 14,900 | 7 | 4 | 5 ± 1 | 0.71 |
| *A. aerophob* a 2 | A2 | *Verongida* | 15,285 | 32 | 13 | 15 ± 2 | 0.84 |
| *A. aerophoba* 3 | A3 | *Verongida* | 17,472 | 21 | 21 | 11 ± 4 | 0.81 |
| *A. damicornis* 1 | D1 | *Halichondrida* | 14,900 | 0 | 0 | – | – |
| *A. damicornis* 2 | D2 | *Halichondrida* | 15,572 | 0 | 0 | – | – |
| *A. damicornis* 3 | D3 | *Halichondrida* | 17,472 | 0 | 0 | – | – |
| *A. verrucosa* 1 | V1 | *Halichondrida* | 13,583 | 0 | 0 | – | – |
| *A. verrucosa* 2 | V2 | *Halichondrida* | 13,701 | 0 | 0 | – | – |
| *A. verrucosa* 3 | V3 | *Halichondrida* | 130,365 | 1 | 1 | 1 ± 0 | 0 |

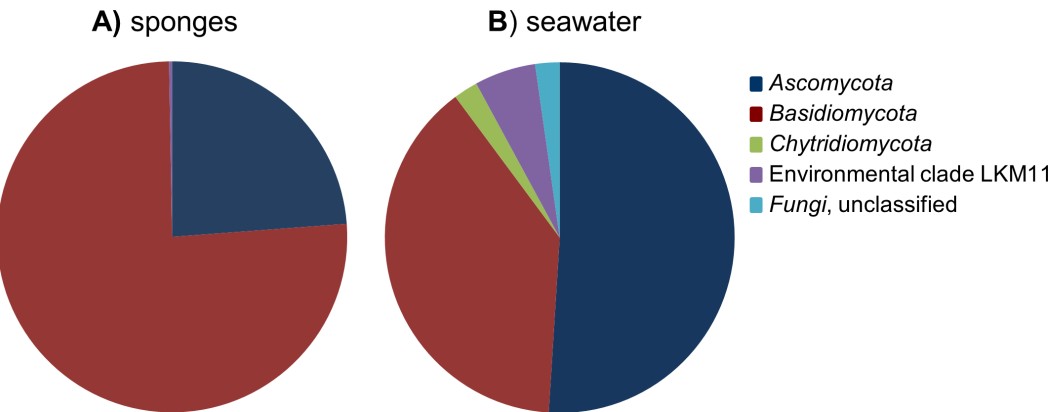

**A) sponges**

**B) seawater**

- ■ *Ascomycota*
- ■ *Basidiomycota*
- ■ *Chytridiomycota*
- ■ Environmental clade LKM11
- ■ *Fungi*, unclassified

**Figure 2** Relative abundance of fungal phyla found in the cumulative sponge and seawater samples.

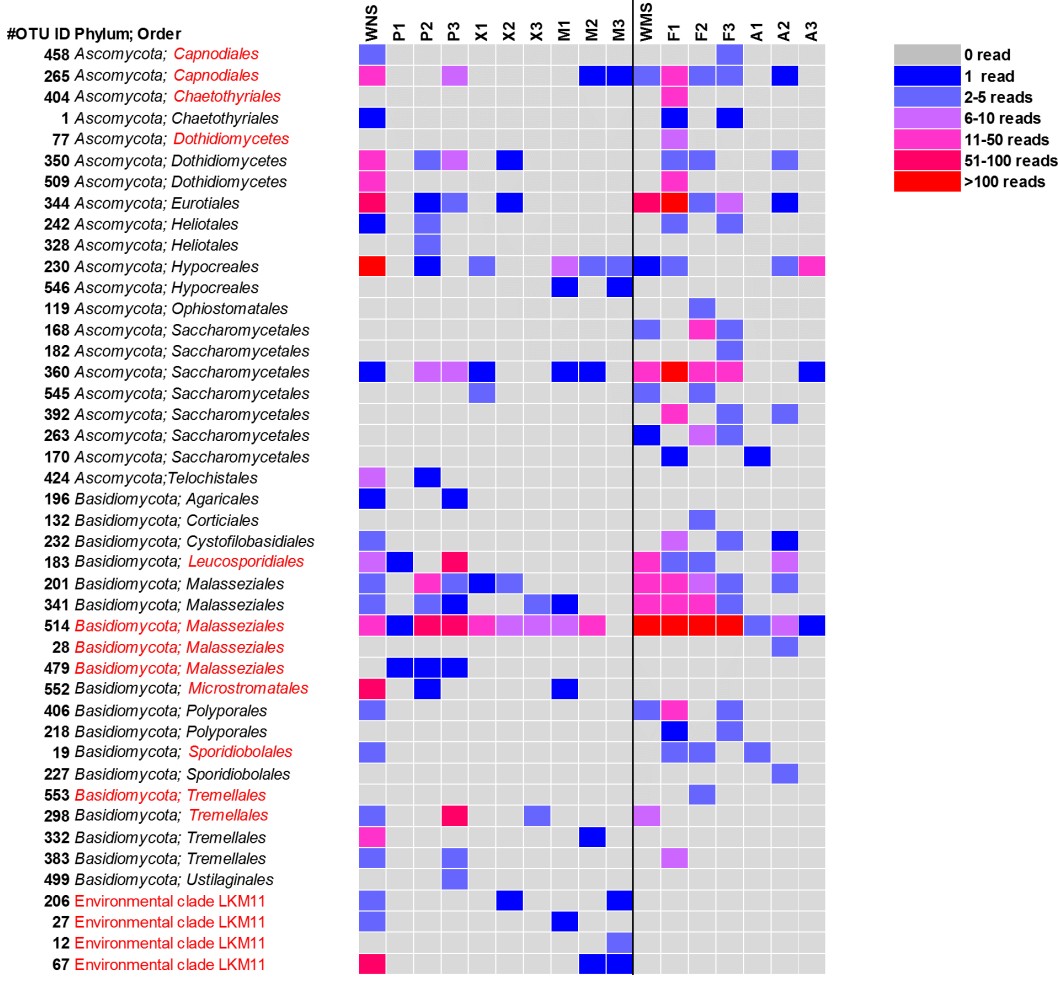

**Figure 3** **Heatmap of fungal OTUs in North Sea sponges and Mediterranean Sea sponges.** OTUs that were only found in seawater and singletons are not shown. Taxonomic affiliation is shown at phylum level and order level whenever possible. Some OTUs (in red) were re-classified based on Bayesian phylogenetic analysis (see also Fig. 4). WNS, North Sea water; P, Halichondria panicea; X, Haliclona xena; M, Suberites massa; WMS, Mediterranean seawater; F, Petrosia ficiformis; A, Aplysina aerophoba. Numbers 1, 2 and 3 refer to different individuals of the sponge species.

reads found in sponges. The large majority (79% of the reads classified as yeasts) of these yeasts was classified to the order *Malasseziales*. OTU514 was the most dominant fungal OTU detected and belonged to this order. It was detected in all sponge species for which fungal reads were obtained (Fig. 3). Other yeasts encountered in sponges belonged to the orders *Saccharomycetales*, *Leucosporidiales*, *Tremellales*, *Cystofilobasidiales*, *Sporidiobolales*, *Ustilaginales*, *Microstromatales* and *Sporidiales* (in order of decreasing relative abundance) (Figs. 3 and 4).

The other 22 fungal OTUs were taxonomically identified as non-yeast fungi and belonged to the orders *Eurotiales*, *Dothidiomycetes*, *Hypocreales*, *Chaetothriales*, *Capnodiales*, *Heliotales*, *Polyporales*, *Ophiostomales*, *Telochistales*, *Agaricales*, fungal clade LKM11, and a number of OTUs belonging to *Ascomycota* and *Basidiomycota* could not be reliably

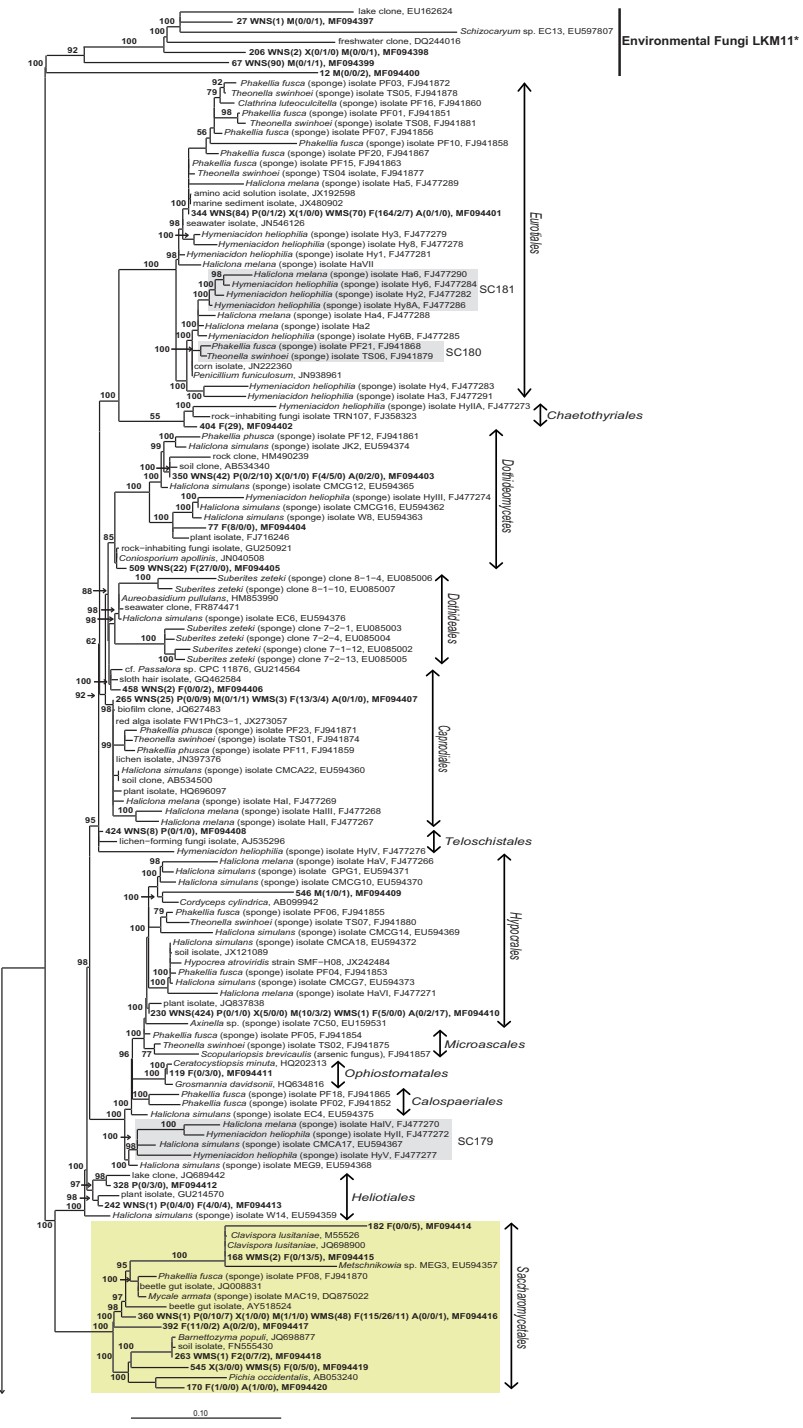

**Figure 4** **Bayesian phylogram of *Ascomycota* OTUs found in sponges based on 18S rRNA gene sequences.** Other sequences included are (i) their nearest neighbors, and (ii) 18S rRNA sequences published by *Simister et al. (2012)* and *Baker et al. (2009)*. Numbers in parentheses after the sample type indication refer to the absolute numbers of reads that were included in the corresponding OTU per sample (34 P(0/0/4) WMS(2) F(4/0/0) means that 4 reads of P3, 2 reads of (continued on next page...)

**Figure 4 (…continued)**
Mediterranean seawater and 4 reads of F1 are included in OTU34). The NCBI genbank accession number is the last descriptor for each branch. Grey boxes indicate sponge-specific clusters (SSC) as defined by *Simister et al. (2012)*. Yellow boxes represent yeasts. Taxonomic groups marked with an asterisk (*) indicate non-formal taxonomic classification (i.e., fungal environmental clade LKM11 is not a phylum, but a group assigned for environmental fungal-like sequences; *Lara, Moreira & López-García, 2010*). The numbers above or below the branches correspond to posterior probability (PP) values of the Bayesian analysis. Nodes with PP values of < 50 are not indicated.

classified at the order level (Fig. 4). The most abundant non-yeast fungal OTU (OTU344) belonged to the order *Eurotiales* and was closely related to fungi previously isolated from other sponges, but was also present in both Mediterranean and North Sea seawater samples.

Sixteen of the 44 fungal OTUs that were detected in sponges were not detected in seawater. However, only two of these OTUs were represented by more than 10 reads (Fig. 3). These are OTU392, a yeast belonging to the order *Saccharomycetales* and OTU404, a fungus belonging to the order *Chaetothyriales*. In addition, several fungal OTUs were found to be present both in seawater and in sponges, but were represented by a higher number of reads in sponges. OTUs that collected substantially more reads in sponges include the most abundant fungal OTU of the dataset, OTU514 (*Malasseziales*). Other yeasts that collected more reads in sponges than in seawater were OTU183 (order *Leucosporidiales*, enriched in *H. panicea*), OTU298 (order *Tremellales*, enriched in *H. panicea*) and OTU360 (order *Saccharomycetales*, enriched in *H. panicea*, *H. xena*, *S. massa*, *P. ficiformis*, *A. aerophoba*). Furthermore, for the non-yeast fungus OTU344 (order *Eurotiales*) more reads were obtained from *P. ficiformis* than from seawater. It should be noted, however, that these 'enriched' fungal OTUs in sponges were also all detected in both North Sea and Mediterranean seawater and that they were often found in multiple sponge species (Fig. 3).

## DISCUSSION

The large majority of 18S rRNA gene sequences that was obtained from sponges were derived from the sponge hosts. In total 2,550 fungal reads were obtained from the sponges investigated (including the reads obtained as singletons). These reads represented less than 0.75% of all the reads that were obtained from the sponges. The number of fungal reads obtained from the sponge samples in our study was similar to, or exceeded, the number of fungal reads reported in other studies that have applied next generation sequencing technology to study fungal diversity in sponges (*Passarini et al., 2015*; *Rodríguez-Marconi et al., 2015*). Therefore, with respect to numbers, these studies are more comparable to past studies that used clone libraries for diversity estimations of bacterial diversity in sponges (*Hentschel et al., 2002*; *Sipkema & Blanch, 2010*) than to studies that apply next generation sequencing technology. That implies that the interpretation and discussion of our results remains mostly limited to qualitative statements as the limited statistical power based on the data did generally not allow quantitative statements. However, one quantitative statement that can be made for the seven sponges studied here is that with the exception of *P. ficiformis* the number of fungal reads was very low in comparison to the number of

sponge reads (2,550 reads vs. 330,884 reads). This is in line with the low relative abundance of fungi in nineteen other sponge species that were assessed by cultivation-independent methods (*Gao et al., 2008*; *He et al., 2014*; *Passarini et al., 2015*; *Rodríguez-Marconi et al., 2015*; *Wang et al., 2014*). The low number of fungal reads also corroborates the lack of microscopic observation of fungi reported from sponge tissues. These low numbers and lack of observations are in sharp contrast to the numbers of bacteria that are present in especially high-microbial-abundance sponges, and which may account for up to 38% of the sponge tissue volume (*Vacelet & Donadey, 1977*).

When analysing the fungal OTUs associated with the sponges, many OTUs were found across different sponge species, across sample types (sponges and seawater) and across geographical regions (Fig. 3). This fungal diversity pattern opposes the bacterial diversity patterns that have been obtained for these sponge species in which every sponge species was observed to have a large fraction of sponge species-specific bacteria, i.e., bacteria found in only one of the sponge species and not or barely found in seawater (*Naim et al., 2014*; *Schmitt, Hentschel & Taylor, 2012*; *Sipkema et al., 2015*). The dominant fungal OTU encountered (OTU514) was a yeast belonging to the order *Malasseziales,* and it was found in all sponges from which fungal reads were obtained. The *Malasseziales* were also found to be the dominant order in cultivation-independent fungal diversity studies of the sponges *Dragmacidon reticulatum* (*Passarini et al., 2015*), *Suberites zeteki* and *Mycale armata* (*Gao et al., 2008*). It is interesting to note that the nearest neighbours of OTU514 all refer to sequences that have been obtained by cultivation-independent methods (Fig. 5 and Table S3). This observation implies that the dominant fungi observed in sponges are different from the ones obtained through cultivation. That suggests that the 'great plate anomaly' that is known to exist for bacteria in sponges also may be true for fungi found in sponges (*Schippers et al., 2012*; *Sipkema et al., 2011*). However, the fact that these yeasts are found in many different sponge species do not yet qualify them as 'sponge-specific' or 'sponge-enriched' fungi as revisited by *Simister et al. (2012)* as the same or highly similar sequences have been obtained from seawater samples in this study and seawater samples from other studies (Fig. 5 and Table S3). Therefore, it is likely that *Malasseziales* are found in sponges because they are filtered from the seawater rather than because they are in a symbiotic relationship with the sponge host.

Without discussing all OTUs that were found in sponges individually, it can be said that 28 of the 44 fungal OTUs that were observed in sponges were also detected in corresponding seawater samples and that these OTUs represented 97.7% of the fungal reads found in sponges (Fig. 3). The two OTUs that represented >10 reads in sponges and were not found in our seawater samples (OTU392 and OTU404) were found most closely related to fungal 18S rRNA gene sequences obtained from anaerobic sludge or rock-inhabiting fungi (Fig. 5 and Table S3). Therefore, it appears that most—if not all—fungal OTUs obtained in this study are not specifically associated to sponges and that they cannot be classified as 'sponge-specific' or 'sponge-enriched' according to the pioneering review by *Simister et al. (2012)*. Three sponge-specific clusters of fungi were proposed by the latter authors (these clusters are indicated in Fig. 4). A remarkable aspect of the fungi in these proposed clusters is that they are all derived from fungal isolates and not from

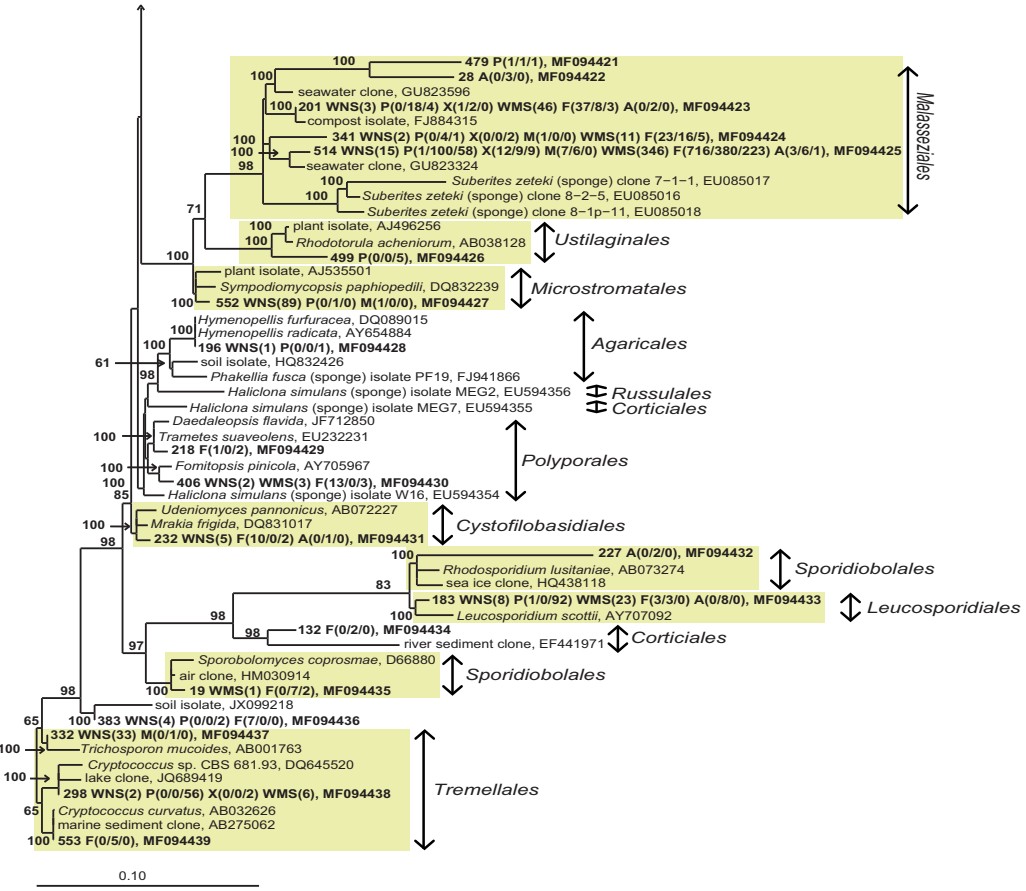

**Figure 5** **Bayesian phylogram of *Basidiomycota* OTUs found in sponges based on 18S rRNA gene sequences.** For additional information see the legend of Fig. 4.

sequences obtained by cultivation-independent means. This is in sharp contrast to their bacterial and archaeal counterparts, as sponge-specific clusters in the domains *Bacteria* and *Archaea* were nearly all obtained from clone libraries (see Supplementary figures in *Simister et al. (2012)*). The state of the art is that we currently know very little about fungi from sponges. Based on the data shown here and other cultivation-independent studies targeting sponge-associated fungi (*Gao et al., 2008*; *He et al., 2014*; *Passarini et al., 2015*; *Rodríguez-Marconi et al., 2015*) there are indications that we know very little because there is little to be known. The combination of low numbers of fungal reads retrieved from sponges with their unspecific nature based on the currently available data point towards merely accidental presence of fungi in sponges. On the other hand there are a few examples that would advocate a more symbiotic relationship between sponges and fungi. The first record is an unidentified encrusting sponge that grows on top of fungi belonging to the genus *Koralionastes* (*Kohlmeyer & Volkmann-Kohlmeyer, 1990*). The second lead stems from the microscopic observations of yeasts that are maternally transmitted in three sponge species belonging to the genus *Chondrilla* (*Maldonado et al., 2005*). A better understanding of potential specific sponge-fungi relationships would benefit from further

exploring these two known relationships. Also for other marine invertebrates, such as corals and tunicates, the nature and consistency of the relationship between the fungi encountered and their hosts have remained unresolved (*Yarden, 2014*). The still ongoing (meta)genomics revolution may be key to shedding light on these aspects (*Scazzocchio, 2014*).

## ACKNOWLEDGEMENTS

We would like to acknowledge Dr. Klaske Schippers and Prof. René Wijffels for sample collection and initial identification of the North Sea sponges and Prof. Maria Uriz for her help with the initial identification of the Mediterranean sponges.

### Funding

Mohd Azrul Naim was funded by Ministry of Higher Education Malaysia (KPT: BS-830623115465) for ''Skim Latihan Akademik IPTA (SLAI)'' program and Detmer Sipkema was funded by European Community's Seventh Framework Program projects MetaExplore (grant agreement number 222625) and BluePharmTrain (grant agreement number 607786). The funders had no role in study design, data collection and analysis, decision to publish, or preparation of the manuscript.

### Grant Disclosures

The following grant information was disclosed by the authors:
Ministry of Higher Education Malaysia: KPT: BS-830623115465.
European Community's Seventh Framework Program projects MetaExplore: 222625.
BluePharmTrain: 607786.

### Competing Interests

Hauke Smidt is an Academic Editor for PeerJ.

### Author Contributions

- Mohd Azrul Naim conceived and designed the experiments, performed the experiments, analyzed the data, contributed reagents/materials/analysis tools, wrote the paper, reviewed drafts of the paper.
- Hauke Smidt conceived and designed the experiments, reviewed drafts of the paper.
- Detmer Sipkema conceived and designed the experiments, analyzed the data, contributed reagents/materials/analysis tools, wrote the paper, prepared figures and/or tables, reviewed drafts of the paper.

### Field Study Permissions

The following information was supplied relating to field study approvals (i.e., approving body and any reference numbers):
For the North Sea sponges:

Sponge collection of the North Sea sponges was approved by the Provincie Zeeland (document number 0501560).

For the Mediterranean sponges:

The collection of sponge samples was conducted in strict accordance with Spanish and European regulations within the rules of the Spanish National Research Council with the approval of the Directorate of Research of the Spanish Government. The study was found exempt from ethics approval by the ethics commission of the University of Barcelona since, according to article 3.1 of the European Union directive (2010/63/UE) from the 22/9/2010, no approval is needed for sponge sacrification, as they are the most primitive animals and lack a nervous system. Moreover, the collected sponges are not listed in CITES.

## DNA Deposition

The following information was supplied regarding the deposition of DNA sequences:

Pyrosequencing data was deposited at the European Bioinformatics Institute with accession numbers ERS225550–ERS225575.

Representative reads for fungal OTUs that were found in sponge samples excluding singletons were deposited at NCBI GenBank with accession numbers MF094397–MF094440.

## Data Availability

The raw data has already been deposited at the European Bioinformatics Institute with accession numbers ERS225550–ERS225575.

Representative reads for fungal OTUs that were found in sponge samples excluding singletons were deposited at NCBI GenBank (MF094397–MF094440) and can also be found in Table S4.

## Supplemental Information

Supplemental information for this article can be found online at http://dx.doi.org/10.7717/peerj.3722#supplemental-information.

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
