# Peer review of "Fungi found in Mediterranean and North Sea sponges: how specific are they?"

_PeerJ, doi:10.7717/peerj.3722_

## Round 0.1 · original submission · Major Revisions

Dear Detmer,

Thank you for your submission to PeerJ your article - Fungi found in Mediterranean and North Sea sponges: How specific are they?
Your manuscript has been reviewed now. While overall the reviewers were very supportive of your discoveries, they have raised a number of concerns, which would require a major revision.

Reviewer comments are shown below and on your article 'Overview' screen.

If you address these changes and resubmit, there's a good chance your article will be accepted (although this isn't guaranteed).

Reviewer 1 ·

Basic reporting

The introduction is well written and provides a suitable motivation for the study. One omission though is the use of the ITS sequence to characterise fungal diversity (not only in terrestrial system, but also in marine environments). I think this molecular marker should also be mentioned and its (dis-)advantages compared to the 18S rRNA marker used here should be discussed (especially in the light of the results found here).

The language and style is of appropriate standard and only needs minor corrections (see below).

Figures are all relevant, but Figure 3 could be expanded (see comment below)

Raw data is deposited in public database, but additional date would be good to provide (see comment below)

Minor corrections:

Line 34: sea fans

Line 289f and line 305: These sentences are redundant and I recommend rephrasing.

Experimental design

This study makes an original contribution as it explore the understudied fungal diversity in sponges. There is a clear research question defined.

The methods are of good technical standard and clearly described. The only concern I have is that two different DNA extraction methods were used for the North Sea and the Med Sea sponges. What was the reason for this?

Validity of the findings

As the richness estimates correlated with the number of fungal reads obtained for each sample (line 198-199), I wonder if the richness estimates are actually meaningful. Given that for some samples only very few reads very obtained (e.g. 3; see Table 1) the richness is likely often been underestimated. To give the reader a better understanding for this issue, I would also recommend that information of community coverage is being presented (e.g. Good’s coverage).

I don’t agree with the conclusion in line 299 that nearly all fungal OTUs found in sponges are also found in seawater. In figure 3 I counted that 13 OTUs were found in sponges, but not in seawater and this is more than 25%. I would suggest rephrasing this statement to be more quantitative. I would also recommend that Figure 3 shows individual replicates as this will provide information, if certain OTUs are consistently associated with certain sample types.

Additional comments

Line 184: It would be interesting for the reader to understand more about what the non-fungi/non-poriferan OTUs are. This does not have to be elaborated in the text, but I would encourage the authors to share their OTU table and taxa assignment per OTU with the community.

·

Basic reporting

Naim et al described the fungal diversity in 7 marine sponge species from Mediterranean and North Sea, using 18S rRNA gene as a phylogenetic marker. This study used amplicon deep sequencing to overcome a major hurdle in symbiotic fungal diversity study. Although useful reads are still limited with only 5 sponge samples have more than 100 reads, the results provided interesting information about variation of fungal diversity in host species level. By comparing to samples from surrounding seawater, authors found no signature of fungal OTU in sponges, suggesting fungi associated with sponge might not be specific. Overall, the science question is clearly stated and the method and statistic are solid, raw data is available through public database. Results are well structured and presented with clean figure and tables. Discussion is comprehensive and self-contained with relevant results. There are very few language issues and the manuscript is easy to read. However, I feel the depth of results can be enhanced by more beta-diversity analysis of the dataset.

Experimental design

1. There are other diversity studies using ITS region for fungal classification purpose. Can author elaborate the rationale for their primer selection in this study and maybe discuss the pros and cons of different primer set in detecting fungal diversity from environmental samples?

2. Were there replicates for seawater samples?

3. I would like to see a PCoA analysis here to visualize distance between samples and draw further conclusion about the specificity and geographic effect on sponge fungal community. Highly variable reads in the sponge samples may be a potential issue for beta-diversity analysis. I suggest pool samples from the same host species together, and normalize them to ca. 50 reads per sample to compare distance between sponges and seawater. Or use jackknife method to estimate the robustness of the rarefied distance matrix.

Validity of the findings

1. It is interesting to observe some sponge species associated with relatively high fungal reads sponge and some with extremely low fungal reads. How many copies of 18S rRNA gene do sponge cells have? And how many for fungal cell? Though not truly reflect the fungal/host cell number, the ratio between fungal and sponge 18S rRNA reads can serve as an index to provide some hints on which sponge species might harbor a fungal microbiota. This might give some clue for subsequence study to focus on certain host species for fungal study, similar to the concept of HMA and LMA sponge in bacterial community studies.
2. Although deep sequencing was applied here, number of fungal reads in each sample is still low and highly variable. I suggest authors provide rarefaction curves for each sponge species and seawater samples as supplemental figures, allow reader have an idea of sequencing depth.
3. I wonder if sponge species with close phylogeny have similar fungal community? It can be done by constructing a tree based on sponge 18S rRNA sequences, and compare it with UPGMA tree generated from distance matrix. Again, PCoA plot can be very helpful here. If fungal communities were separated by Mediterranean Sea samples and North Sea samples, this would suggest that geographic location is the major drive in shaping fungal community. Otherwise, if samples were group based on host phylogenetic relatedness, it would indicate host effect in shaping fungal community, which can serve as an evidence to answer the question: how specific are they?
4. For figure 4A and 4B, I suggest authors deposit the representative sequences from each fungal OTU listed in the phylogenetic tree to public database, then provide an accession number for each OTU. This will make subsequent comparative studies a bit easier for science community.

Additional comments

After first appearance in the text, species names should be abbreviated. Please check this issue for consistence throughout the text.
Some minor grammar issue:
Line 47: I suggest a comma before “knowledge of” for easier reading.
Line 73: I feel “i.e.” is not necessary here.

Reviewer 3 ·

Basic reporting

no comment

Experimental design

This manuscript is a work with previously published data, not being published the data presented contrary to the scope of this periodical:

(Mohd Azrul Naim, Hauke Smidt and Detmer Sipkema. Chapter 3: Molecular analysis of sponge- associated fungi In: Exploring Microbial diversity of marine sponges by culture-dependent and molecular approach Mohd Azrul Naim Mohamad, 2015, Wageningen University, 222 pg. ISBN 978-94-6257-286-7)

1- Abstract:
- Therefore our data, supported by currently available data, point into the direction of merely accidental presence of fungi in sponges and do not support the existence of a sponge-specific fungal community.

The affirmation of the presence of otus corresponding to fungic groups as merely accidental, contradicts all references and studies with fungal communities associated to sponges. The use of specific non-fungal tools compromised the achievement of such results, negatively impacting the obtained results, but did not support such a statement
2 - References: et al., is not italic, references is out of the rules of the journal; p.e. (l. 12 Haeksworth, 2001; Muller & Schmit, 2007);.
3 - L.30 – L.36 niches life of the fungi
4- Goal: The aim of this study was to assess the diversity and specificity of sponge-associated fungi. Seven shallow water sponge species from two different regions, i.e. the North Sea and the Mediterranean Sea, were sampled to identify host specificity of the associated fungal communities and the impact of geography on community structuring.

Although the authors worked with different species from different environments (North Sea and Mediterranean Sea), the differences between the physico-chemical characteristics and the physiological diversity were not explored. They have not made clear on what can be influencing the distribution of diversity. What is the sponge community structure and the microbial expected for these exploited environments?

5- Material and methods
I suggest authors, to build one table with described with sponge species collected from different locations and taxonomy identification.

6- The authors not specific what kind piece of the sponge was used to extraction protocol, for example: pinacoderm and choanderm. How depth the sponges were collected? were collected within a 25-m radium at depths of 0.5–3 m.

7- L. 85-86. Improve the technical specification to process the seawater samples: p.e.
The seawater samples were pooled (5 – 10 L in total) and filtered on xx-cm-diameter micro-pore (0.22 μm) polycarbonate filters (GE Osmonics, Minnetonka, MN, USA). Each filter was then kept in 20 mL RNA Later – 80 °C until further treatment.

8- DNA extraction and PCR amplification
The authors used two different extraction protocols L.90 and L.91, Are there one specifics material?

9- L.94, what size the filter were cut?

10- L.98, The universal fungal primers used was FF390 and FR1 (Vainio & Hantula 2000). - PCR primers play a crucial role in the molecular assessment of environmental microbes, thus the evaluation of primer specificity and efficiency is necessary. The specificity of the primer pairs is vital and allows selective or enriching amplification of fungal rRNA genes from environmental DNA (Pang & Mitchell, 2005). In general there are one large combined primers pair for studies of fungal communities, with a robust framework such Small subunit rRNA (SSU rRNA), internal transcription space (ITS) and large subunit rRNA (LSU rRNA) also used as a phylogenetic marker, as reference by Schoch et al., 2012). Gao et al. (2008) tested over 10 primer sets on Hawaiian sponge Suberites zeteki, including three ITS primer sets. According to their observations, ITS1/ITS4 was the only suitable primer set, whereas ITS1/ITS2 offered no fungal sequences. Hence, considering the inference of sponge ITS, the fungal ITS region may not be a universal barcode for studying fungal diversity in sponges. Among seven primer sets targeting SSU and LSU, EF4/fung5 yielded the best results, although this primer set had lower coverage in Basidiomycota compared to EF4/EF3 (Smit et al., 1999).
However, in this study the authors were used FF390 and FR1, not only specific to fungal community but, also previously been found to amplify 18S rRNA genes of other non-fungal eukaryotes such as Choanoflagellida, Mesomycetozoa, Cnidaria and Porifera sequences (Hoshino & Morimoto, 2010, Chemidlin Prévost-Bouré, et al., 2011). Therefore, the best approaches to access fungal communities were not used.
A very recent pyrosequencing survey of fungal and protistan diversity in sponge using eukaryotic universal primers (3NDf and euk_v4_R2) showed that amplicons of sponge 18S r RNA gene composed over 80% of total reads, which made it an inefficient way to target fungal diversity (He et al. 2014). Together with previous researches, this results showed that when dealing with different sponge species, the specificity of primers may vary and the inference of sponges rRNA genes cannot be ignored in which case, the coverage of primers might not be the prior concern.
According to the recent review from Lindahl et al. 2013, fungal 18S rRNA had very limited phylogenetic resolution at lower ranks and identical sequences could be found in hundreds of other species across entire fungi, which meant it was not always accurate to classify a fungi sequence based on a perfect Blast hit.
I suggest the authors, in this study applied a composition-based method (naïve Bayes classifier) to address the taxonomic assignment (Wang et al. 2007; Liu et al. 2012).

11- L.105, The final PCR mixture: replace 50μl by 50 μL;

12- L.110 replace 1 μg/ml by 1 μg/mL;

13- Sequences and species richness analysis
L.118 Caporaso et al., 2010 replace by Caporaso et al., 2010

14- OTU correlation : Is there a correlation exists between OTUs and other attributes of sample like pH or other environmental conditions. 


Differences in taxa at various levels of taxonomy? 

What are the top most OTUs (or species)? 

What are the rare OTUs present? 

OTU correlation : Is there a correlation exists between OTUs and other 
attributes of sample like pH or other environmental conditions. 

Sample Size / Power Calculation 

Differences between 2 or more groups : T-test /ANOVA/PERMANOVA 

Multiple testing Correction 

Correlation with other attributes 
Fungal Taxa distribution (Family, genus level): percentage distribution of fungi taxa
Graphs : Taxa distribution (Heatmaps), PCoA plots, Diversity plots etc. 

E.g. high concentrations of nitrate and nitrite, no light for CO2 fixation by photosynthesis, therefore N and C related metabolism were particularly were particularly highlighted.
What the author propose eukaryotic symbionts as well as their relationships with sponge host were suggested…

Validity of the findings

Totally unfounded discussion, with unfounded assertions, with contradictions in several paragraphs.

The work is about analysis of diversity, however, the indices offered for such studies were not explored. Although we have accessed the restricted number of OTUs, but the same did not limit the metric studies of diversity of the fungic groups accessed, such as, alpha diversity and riches. The authors did not present the indices of riches, relativities, alpha diversities

16- L. 267 Failure to observe fungal cells in the sponge tissue does not exclude the presence of fungal microorganisms living in association, since fastidious microorganism cultivation and access techniques are limited. Rewrite this paragraph

17- L.310 Although a few but robust paper discussing the presence of fungal communities associated with sponges, the authors used data from bacterial communities to broaden their discussion. I suggest here the authors consult references of studies of the fungal communities to resolve the possible main points of the studies.

18- L. 317 the author .. “there are indications that we know very little because there is little to be known.” Is completitly wrong way, it despite, the need tools more complexy and combined with complete with different approaches

19 -Including in discussion:

So far it has not been determined whether the symbiosis of sponge and fungi is a parasitic one. However, there are several lines of evidence that can give us some ideas about the fungal roles in the sponge-fungi symbioses. On one hand, sponge-associated fungi display diverse biological activities, which make them the most prolific sources for bioactive compounds. Sponge-associated might be involved in the chemical defense of the sponge host. On the other hand, marine fungi are thought to be major contributors to the decomposition of organic matter (e.g. chitin, lignin) in coastal and marine surface environments. Hence, the roles of fungi in the nutrient cycling within sponges can not be ignored. Last but not least, fungi in water columns are of importance within the microbial food web in the coastal oceans in many ways, eg. Controlling energy flow, regulating food web dynamics, and with spores serving as food for zooplankton.

- Hence, the ecological roles of sponge-associated fungi should not be neglected/underestimated. It is worth exploring the functional gene diversity of sponge-associated fungi in future studies. Our understanding of the function of marine fungi is still quite limited

---

## Round 0.2 · Minor Revisions

Your manuscript requires just a few remaining minor changes. Please consider all comments. modify your manuscript accordingly and resubmit it.

Reviewer 1 ·

Basic reporting

The introduction is well written and provides a suitable motivation for the study. One omission though is the use of the ITS sequence to characterise fungal diversity (not only in terrestrial system, but also in marine environments). I think this molecular marker should also be mentioned and its (dis-)advantages compared to the 18S rRNA marker used here should be discussed (especially in the light of the results found here).

>>A We have now incorporated the use of ITS sequences to assess fungal diversity in sponges in the text at points where we previously only mentioned the 18S rRNA gene. For this part, we also introduced an additional reference by Jin et al.
In a number of papers the success of multiple primer sets for ITS and 18S rRNA genes have been compared for assessing sponge-associated fungal diversity (Gao et al., 2008; Jin et al., 2014). Although ITS appears to be a better marker for taxonomic assignment of fungi, both genes seem to be non-discriminative between sponge sequences and fungal sequences. Therefore, we only chose to use one marker gene for this study with the idea to overcome this non-discriminative aspect by sheer numbers. That was only moderately successful as more fungal reads than we observed would have provided more evidence and current illumina sequencing methods will lead to deeper insights in fungal communities in sponges. For this reason we have been careful not to overinterpret our data and only major OTUs are discussed.

Second round comment:
Thanks for this additional information and consideration. I am happy with this aspect now.

The language and style is of appropriate standard and only needs minor corrections (see below).

Figures are all relevant, but Figure 3 could be expanded (see comment below)
>>A Figure 3 has been adapted according to the reviewer’s suggestion

Second round comment:
Thanks for modifying Figure 3. Only additional comment here it to provide an explanation for the sample code in the figure legend.

Raw data is deposited in public database, but additional date would be good to provide (see comment below)
>>A The data the reviewer is asking for is mentioned in more detail below and is available at:
http://edepot.wur.nl/339792
In addition, genbank accession numbers have been generated for representative sequences of the 44 OTUs analyzed in greater detail (see comment reviewer2) to make these partial 18S rRNA sequences more easily available for the scientific community.

Second round comment:
Thanks.

Minor corrections:

Line 34: sea fans
>>A “sea fan” was changed to “sea fans”

Second round comment:
Thanks.

Line 289f and line 305: These sentences are redundant and I recommend rephrasing.
>>A The text was revised by removing two sections:
1. “This higher number of reads retrieved from sponges is noteworthy considering the fact that most 18S rRNA gene reads obtained from sponges were classified as Porifera and therefore reduced the number of fungal reads that were obtained from sponge samples.”
2. “Both in H. panicea and in P. ficiformis the number of reads of this OTU in an individual sample was more than the double of the reads obtained from the corresponding seawater samples.”

Second round comment:
Thanks. Reads much better now.

Experimental design

This study makes an original contribution as it explore the understudied fungal diversity in sponges. There is a clear research question defined.

The methods are of good technical standard and clearly described. The only concern I have is that two different DNA extraction methods were used for the North Sea and the Med Sea sponges. What was the reason for this?

>>A The reviewer is correct with the underlying assumption that different DNA extraction methods may lead to different microbial communities and that preferably all samples should be extracted according to the same method. In this case this did not happen mainly due to the different moments in time that DNA extractions were done: in 2008 in The Netherlands and in 2012 in Spain. Interestingly, and a bit to my surprise, many OTUs were found to be shared among the samples from different geographical origin, which somehow indicates that the different DNA extraction methods used did not have a major impact on the fungal DNA extracted.

Second round comment:
Thanks for this additional explanation.

Validity of the findings

As the richness estimates correlated with the number of fungal reads obtained for each sample (line 198-199), I wonder if the richness estimates are actually meaningful. Given that for some samples only very few reads very obtained (e.g. 3; see Table 1) the richness is likely often been underestimated. To give the reader a better understanding for this issue, I would also recommend that information of community coverage is being presented (e.g. Good’s coverage).

>>A Indeed, for a number of samples the richness estimations are not meaningful as for those samples the number of fungal reads are too low. We also acknowledge this in the manuscript. For example in the discussion we state that “interpretation and discussion of our results remains mostly limited to qualitative statements as the limited statistical power based on the data did generally not allow quantitative statements”.
For the revised version we did now include Good’s coverage for all samples to Table 1 and added corresponding text to the materials and methods section.

Second round comment:
Thanks. With the Good’s coverage provided, the reader can now much easier assess which samples have been sufficiently sampled.

I don’t agree with the conclusion in line 299 that nearly all fungal OTUs found in sponges are also found in seawater. In figure 3 I counted that 13 OTUs were found in sponges, but not in seawater and this is more than 25%. I would suggest rephrasing this statement to be more quantitative. I would also recommend that Figure 3 shows individual replicates as this will provide information, if certain OTUs are consistently associated with certain sample types.

>>A Indeed (actually) 16 OTUs were found in sponges, but not in seawater (Fig. 3). They represent a substantial fraction of the fungal sponge OTUs analyzed (36%), but a minor part of the cumulative reads of these OTUs (2.3%). However, we have rephrased the text here to more specifically report this finding.
In addition, we have adapted Fig. 3 and it now shows the individual samples as the reviewer suggested.
Second round comment:
Thanks. This is clearer now.

·

Basic reporting

No comment

Experimental design

No comment

Validity of the findings

No comment

Additional comments

The revised ms by Naim et al. is greatly improved, and the response addressed most of my concerns. I see some minor changes to be done to clarify some issues (see below).
Figure legend for supplementary Figure 1 (PCoA plot) is missing.
What the PCoA would look like when two seawater samples were in the same plot? That may give readers a sense how similar the two fungal communities is in different environment. Alternative, a dendrogram on top of the heatmap (Figure3) would serve the same purpose.

---

## Round 0.3 · accepted · Accept

Dear Detmer,

Thank you for your submission to PeerJ which I am happy to Accept.

Congratulations again, and thank you for your submission.

With kind regards,
Marina Kalyuzhnaya
Academic Editor, PeerJ